# Acetylation-Mimic Mutation of TRIM28-Lys304 to Gln Attenuates the Interaction with KRAB-Zinc-Finger Proteins and Affects Gene Expression in Leukemic K562 Cells

**DOI:** 10.3390/ijms24129830

**Published:** 2023-06-06

**Authors:** Yao-Jen Chang, Steven Lin, Zhi-Fu Kang, Bin-Jon Shen, Wen-Hai Tsai, Wen-Ching Chen, Hsin-Pin Lu, Yu-Lun Su, Shu-Jen Chou, Shu-Yu Lin, Sheng-Wei Lin, Yin-Jung Huang, Hsin-Hui Wang, Ching-Jin Chang

**Affiliations:** 1Institute of Biological Chemistry, Academia Sinica, Taipei 11529, Taiwan; ntugeorge@gmail.com (Y.-J.C.); stevenlin@gate.sinica.edu.tw (S.L.); shenbj1967@gmail.com (B.-J.S.); s89401wh@gmail.com (W.-H.T.); f94b46010@ntu.edu.tw (Y.-L.S.); fish1025@gate.sinica.edu.tw (S.-Y.L.); sanway@gate.sinica.edu.tw (S.-W.L.); 2Graduate Institute of Biochemical Sciences, College of Life Science, National Taiwan University, Taipei 10617, Taiwan; tomlichking@gmail.com (Z.-F.K.); a0972718683@gmail.com (W.-C.C.); phoebe1620@gmail.com (H.-P.L.); 3Institute of Plant and Microbial Biology, Academia Sinica, Taipei 11529, Taiwan; sjchou@gate.sinica.edu.tw; 4Department of Pediatrics, Division of Pediatric Immunology and Nephrology, Taipei Veterans General Hospital, Taipei 11217, Taiwan; dlnagb0857@gmail.com (Y.-J.H.); hhwang@vghtpe.gov.tw (H.-H.W.); 5Department of Pediatrics, Faculty of Medicine, School of Medicine, National Yang Ming Chiao Tung University, Taipei 112304, Taiwan; 6Institute of Emergency and Critical Care Medicine, School of Medicine, National Yang Ming Chiao Tung University, Taipei 112304, Taiwan

**Keywords:** acetylation, CRISPR/Cas9, TRIM28, KRAB-ZNF, K562

## Abstract

TRIM28/KAP1/TIF1β is a crucial epigenetic modifier. Genetic ablation of *trim28* is embryonic lethal, although RNAi-mediated knockdown in somatic cells yields viable cells. Reduction in TRIM28 abundance at the cellular or organismal level results in polyphenism. Posttranslational modifications such as phosphorylation and sumoylation have been shown to regulate TRIM28 activity. Moreover, several lysine residues of TRIM28 are subject to acetylation, but how acetylation of TRIM28 affects its functions remains poorly understood. Here, we report that, compared with wild-type TRIM28, the acetylation-mimic mutant TRIM28-K304Q has an altered interaction with Krüppel-associated box zinc-finger proteins (KRAB-ZNFs). The *TRIM28*-K304Q knock-in cells were created in K562 erythroleukemia cells by CRISPR-Cas9 (Clustered regularly interspaced short palindromic repeats/CRISPR-associated protein nuclease 9) gene editing method. Transcriptome analysis revealed that *TRIM28*-K304Q and *TRIM28* knockout K562 cells had similar global gene expression profiles, yet the profiles differed considerably from wild-type K562 cells. The expression levels of embryonic-related globin gene and a platelet cell marker integrin-beta 3 were increased in *TRIM28*-K304Q mutant cells, indicating the induction of differentiation. In addition to the differentiation-related genes, many zinc-finger-proteins genes and imprinting genes were activated in *TRIM28*-K304Q cells; they were inhibited by wild-type TRIM28 via binding with KRAB-ZNFs. These results suggest that acetylation/deacetylation of K304 in TRIM28 constitutes a switch for regulating its interaction with KRAB-ZNFs and alters the gene regulation as demonstrated by the acetylation mimic TRIM28-K304Q.

## 1. Introduction

The evolutionarily conserved, vertebrate-specific tripartite motif protein 28 (TRIM28), also known as KRAB-associated protein 1 (KAP1) and transcription intermediary factor-β (TIF1β), is an essential developmental regulator—specifically, a universal transcriptional corepressor for the KRAB-ZNFs, a large family of transcriptional repressors in vertebrates. The mammalian genome encodes >350 KRAB-ZNFs [1,2,3] involved in growth and differentiation [4,5], the majority of which consists of sequence-specific DNA-binding C2H2-type zinc-finger modules and a KRAB domain that recruits TRIM28.

TRIM28 is composed of an N-terminal unstructured region (amino acid residues 1–62), an RBCC (RING, B1 and B2 boxes, coiled-coil domain) motif, 2 unstructured central regions (residues 406–623, 673–696), a PHD finger and bromodomain, and a C-terminal unstructured region (residues 813–835). The RBCC facilitates the homodimerization of TRIM28 and specifically interacts with the KRAB domain of ZNFs [6,7]. The crystal structure and site-directed mutagenesis demonstrated that the KRAB domain was bound to the middle of the coiled-coil region (residues 294–321) [6,7]. TRIM28 serves as a scaffold for corepressor proteins, including DNA methyltransferases (DNMTs), the Nucleosome Remodeling and Deacetylase (NuRD) complex, the H3K9me3 methyltransferase SETDB1, G9a (EHMT2), and HP1, and this holo-complex contributes to histone modifications and heterochromatin formation to maintain gene silencing [8,9,10,11,12]. Through KRAB-ZNFs and other transcription factors (e.g., OCT4, p53, STAT1, STAT3, etc.), TRIM28 can be targeted to chromatin [13]. The genome-wide analysis presented that TRIM28 and KRAB-ZNFs can target 3’exon of *KRAB-ZNFs* genes and mediate long-range auto-regulatory transcriptional repression [14,15,16,17]. Moreover, KRAB-ZNFs recruit TRIM28 corepressor complexes to specific transposable elements (TEs) to silence their expression [18].

Posttranslational phosphorylation and sumoylation of TRIM28 contribute to the regulation of its functions [19,20,21,22,23,24,25,26,27]. The phosphorylation at Ser473 and Ser824 is well studied. The RING domain contributes to TRIM28 auto-SUMOylation. The acetylation of lysine residues in histones, specific transcription factors, chromatin remodelers, metabolic enzymes, and many regulatory proteins is another widely recognized example of a posttranslational modification that significantly alters protein function and cellular regulation [28]. Several acetylation residues in the coiled-coil and bromo regions of TRIM28 have been identified [29]. Although the acetylation of histones is widely known to play critical roles in regulating gene expression [30,31,32], the impact of TRIM28 acetylation on its activity remains poorly understood. To address this dearth of knowledge, we targeted some acetylation sites in the coiled-coil domain. Upon mutation of the individual mouse Trim28 lysines (K) to acetylation-mimic glutamine (Q) or acetylation-defective arginine (R), only K305Q was compromised to the interaction with KRAB-ZNFs. The gene expression ensemble of the erythroleukemic cell line K562 harboring an endogenous mutation of K304Q differed considerably from that of the parent K562 line. We propose that TRIM28-K304Q could be used as an acetylation mimic for systematic investigation of the potential effects of acetylation/deacetylation on phenotypic changes at both the cellular and organismal levels in a spatiotemporal manner.

## 2. Results

### 2.1. The Acetylation Mimic TRIM28-K304Q Has a Relatively Weak Interaction with the KRAB Domain

Acetylation of human TRIM28-K266, -K304, and -K340 (equivalent to mouse Trim28-K267, -305, and -K341) in the RBCC region has been identified by mass spectrometry (MS) in human acute myeloid leukemia cell line MV4-11 [29]. We also found several lysine residues in the coiled-coil domain were acetylated by MS analysis of immunoprecipitated TRIM28 in leukemia K562 cells, including K254, K261, K289, K304, K319, K366, and K377 (Appendix A). We produced the FLAG-tagged acetylation-mimicking mutants of mouse Trim28, namely K255Q, K267Q, K290Q, and K305Q, as well as acetylation-defective mutants K255R, K267R, K290R, and K305R. These constructs were expressed in TRIM28 knockout human leukemia K562 cells [33] to test their interaction with Gal4 DNA-binding domain (Gal4^DBD^)-KRAB. In an immunoprecipitation analysis, only K305Q had impaired interaction with recombinant Gal4^DBD^-KRAB (Figure 1a). Then, the HA-tagged human TRIM28 wild-type (WT), K304Q, and K304R mutants were generated to examine the KRAB interaction (Figure 1b). The K304Q mutant also displayed lower interaction with KRAB domain compared to WT and K304R mutant. Furthermore, we performed EMSA (Electrophoretic mobility shift assay) to monitor the interaction between Trim28 and DNA-bound Gal4^DBD^-KRAB. The results indicated that WT (W) Trim28 associated with GAL4^DBD^-KRAB bound to GAL4-binding elements, and Trim28-K305Q declined this association (Figure 1c). The functional luciferase assay demonstrated that K304Q mutant had decreased transcription-suppressor activity (Figure 1d, upper panel). Western blotting revealed similar protein abundance in each reaction (Figure 1d, lower panel). It is likely that both the K305Q homodimer and K304-K305Q heterodimer contributed to the decreased binding to the KRAB domain. In 293T cells, wild-type HA-Trim28 was co-expressed with wild-type or FLAG-K267R, -K267Q, -K305R, or -K305Q. The protein complexes brought down by HA beads were detected by anti-FLAG (Appendix A). These results indicated that these mutants can associate with wild-type Trim28. Furthermore, 293T cells were transfected with mouse FLAG-Trim28-K305Q, followed by IP (Immunoprecipitation) and LC-MS/MS analysis, which identified a specific peptide (residues 32–69) from human TRIM28 (Appendix A). These results suggested that the acetylation mimic Trim28-K305Q had decreased interaction with the KRAB domain, both physically and functionally, but did not change dimer formation activity.

### 2.2. Generation of TRIM28-K304Q K562 Cells by CRISPR/Cas9-Mediated Genomic Editing

To investigate the function of TRIM28 acetylation, human *TRIM28*-K304Q was knocked into K562 cells (Figure 2a). A SacI restriction site was inserted for genome examination. Droplet digital PCR further checked several possible clones. Clones Q1 and Q2 each had three K304Q alleles (Appendix A). After confirmation via Sanger sequencing and Western blotting, only Q1 was the correct K304Q knock-in (KI) clone, and Q2 lacked TRIM28 expression owing to the deletion of specific sequences (Figure 2b, Appendix A). The TRIM28 protein from Q1 was immunoprecipitated with anti-TRIM28-N (Section 4) followed by LC-MS/MS. The TRIM28 sequence coverage was 84%, and the K304Q mutation was confirmed (Figure 2c,d). The cell proliferation assay with wild-type, K304Q, and *TRIM28*-KO K562 cells revealed that K304Q cells grew slower than the wild-type cells, and *TRIM28*-KO cells grew very slowly (Appendix A). The sensitivity of these cells to histone deacetylase (HDAC) inhibitor SAHA (suberoylanilide hydroxamic acid, Vorinostat) and tyrosine kinase inhibitor imatinib were analyzed by cell viability analysis, which revealed that wild-type cells were more sensitive to imatinib than the other two mutant cells. These results were consistent with the relatively higher proliferation rate of wild-type cells than the mutants. *TRIM28*-KO cells were more sensitive to SAHA than wild-type or K304Q cells (Appendix A). *TRIM28*-K304Q cells became attached cells, whereas the wild-type cells remained unattached (Appendix A).

### 2.3. RNA-Sequencing of Wild-Type, TRIM28-K304Q KI, and TRIM28-KO K562 Cells

To assess global gene expression in wild-type, *TRIM28*-K304Q KI, and *TRIM28*-KO cells, we performed RNA sequencing (RNA-seq). The K304Q and KO cells had similar gene expression profiles. However, each profile differed from that of wild-type cells (Figure 3a). The results from EdgeR analysis were further analyzed to determine genes with significant differential expression according to the criteria of fold change greater than 2 and q value less than 0.05. The number of up- and down-regulated genes are summarized in Figure 3b. In K304Q cells, 274 genes were upregulated and 285 downregulated compared with wild-type cells. A gene ontology (GO) analysis of K304Q and KO cells revealed the upregulation of DNA-binding proteins such as ZNFs (Figure 3c and Appendix A). Interestingly, the megakaryocytic repressor IKAROS family transcription factors *IKZF2* and *IKZF3* and the paternally imprinted genes *PEG3* and *DLK1* were upregulated, and the embryonic and fetal globin repressor *SOX6*, the maternally imprinted gene *MEG3*, and melanoma antigen family members *MAGEC1, MAGEC2, MAGEB1*, and *MAGEA1* were downregulated (Appendix A). Previously, we had demonstrated that both *SOX6* and *MAGEC2* were downregulated in *TRIM28*-KO cells, which were analyzed by cDNA microarray [33]. To further verify the up- or downregulation of these genes, we performed RT-qPCR (Figure 3d), which confirmed the changes in the expression of specific genes identified by RNA-seq of *TRIM28*-K304Q cells. The imprinting gene, which encodes the long noncoding RNA *H19*, was significantly upregulated in *TRIM28*-K304Q cells, although *H19* was not listed in RNA-seq data. The *TRIM28*-K304Q-induced perturbation of ZNF expression highlights the essential roles of dynamic regulation of acetylation of TRIM28 (as exemplified by the acetylation mimic TRIM28-K304Q) on gene expression. The trends of gene expression levels in *TRIM28*-KO cells were similar to in *TRIM28*-K304Q cells (Appendix A). For example, the ZNFs were highly upregulated in *TRIM28*-KO cells.

KEGG pathway analysis revealed significant changes in gene expression of pathways such as platelet activation, thyroid hormone signaling, and miRNA**s** in cancer and PI3K/AKT signaling (Figure 4a). Integrin subunit β3 (ITGB3) is involved in these pathways. Integrins are transmembrane heterodimeric receptors that bind the extracellular matrix to transduce the intracellular signaling [34]. ITGB3 is commonly expressed on platelets, acting as a receptor for fibrinogen and fibronectin [35]. We verified the upregulation of its mRNA and protein in *TRIM28*-K304Q and *TRIM28*-KO cells (Figure 4b). Fibronectin was also highly upregulated in K304Q cells but not in *TRIM28*-KO cells (Figure 4b), which might explain the attached phenotype of *TRIM28*-K304Q cells (Appendix A). We also detected the induction of epsilon (*HBE*) and gamma (*HBG*) globin genes in K304Q cells (Figure 4c), which was increased in *TRIM28*-KO cells [33].

### 2.4. HDAC Inhibitor SAHA Induces Gene Expression via TRIM28 Acetylation

We further explore whether the TRIM28-K304Q-mediated gene expression was correlated to TRIM28 acetylation. At first, the wild-type K562 cells were treated with HDAC inhibitor SAHA for 72 h, and RNAs were isolated for RT-qPCR. As shown in Figure 5a, the mRNA expression levels of *ZNF568, ZNF667, ITGB3*, and *HBE* were significantly induced by SAHA. Moreover, we performed ChIP-qPCR to demonstrate the enrichment of TRIM28 on these genes. The results showed that compared to wild-type TRIM28, the TRIM28-K304Q had lower enrichment on the 3’ exon of *ZNF568* and *ZNF667* genes, *ITGB3* and *HBE*, and SAHA treatment decreased the TRIM28 enrichment (Figure 5b). To monitor the K304 acetylation, we generate the anti-acetyl-K304 antibody (Section 4). We found the acetylation signals of TRIM28 were slightly increased during SAHA treatment (from 1.27- to 2.22-fold) and were decreased to basal under the treatment with a HAT inhibitor (Figure 5c). The high signal of acetyl-K304 in K304Q proteins might be due to the recognition of non-specific acetyl residues, and it needs to be further confirmed. Taken together, TRIM28 can respond to the protein acetylation signal to upregulate some of KRAB-ZNFs gene expression via acetylationat K304.

### 2.5. Dynamic Interaction between TRIM28 and KRAB-ZNFs to Regulate Gene Expression: ZNF558 and ZNF445

The above results suggest that the TRIM28-K304Q mutant alters interaction with KRAB-ZNFs to modulate gene expression. Therefore, we performed IP using anti-TRIM28-N-conjugated protein A beads followed by MS analysis to compare the differential interacting proteins between wild-type and TRIM28-K304Q. The bead-bound TRIM28 or TRIM28-K304Q was washed and eluted with antibody-generated peptide, and MS was performed to analyze each eluent. The whole-interacting proteins were listed in Appendix A for wild-type TRIM28 and Appendix A for TRIM28-K304Q. Most TRIM28-interacting proteins were ZNFs; among them, 88 interacted with wild-type, 54 interacted with both wild-type and TRIM28-K304Q, and 7 interacted with TRIM28-K304Q (Table 1). We verified the interaction between TRIM28 and ZNF558, and ZNF785 by IP-Western blotting (Figure 6a). The TRIM28-KO cells were cotransfected with expression plasmids of HA-tagged TRIM28, including wild-type, K304Q, and K304R proteins, and Myc-DDK-tagged ZNF558 and ZNF785 (Figure 6b). The presented results confirmed that lower amounts of ZNF558 and ZNF785 were precipitated by TRIM28-K304Q mutant compared to wild-type and K304R mutant. The average of three independent experiments also showed this trend. Moreover, we proposed that wild-type TRIM28 interacting with these ZNFs to repress targeted gene expression, and acetylation-mimic TRIM28-K304Q impacted this functional effect. Thus, we knocked down ZNF558 in wild-type K562 cells and monitor the RNA expression levels of ZNF568, ITGB3, and LINE1 which was a known target of ZNF558 [36] (Figure 6c). Knockdown of ZNF558 resulted in the upregulation of ZNF568, ITGB3, and LINE.

To further understanding how the interaction between TRIM28 and KRAB-ZNFs affects gene expression in K562 cells, we investigated the TRIM28-ZNF445-H19 axis. ZNF445 is a KRAB-ZNF (Figure 7a), which was reported in modulating *H19* expression [37]. We performed ChIP using anti-TRIM28-N and anti-ZNF445, and TRIM28 and ZNF445 were enriched at the *H19* promoter in wild-type K562 cells but not in K304Q mutant cells (Figure 7b). Although ZNF445 was detected in both wild-type and K304Q TRIM28 associated proteins in the Table 1, Co-IP demonstrated that TRIM28-K304Q interacted weakly with ZNF445 (Figure 7c). Knockdown of ZNF445 in wild-type K562 cells activated H19 expression (Figure 7d) and decreased the enrichment of TRIM28 at the *H19* promoter (Figure 7e). These results suggested that the association of TRIM28 with ZNF445 is at least partially responsible for repressing *H19* expression and that the weakened interaction between TRIM28-K304Q and ZNF445 can derepress *H19* expression.

## 3. Discussion

Lysine acetylation is a reversible posttranslational modification of proteins that may regulate their interactions with other macromolecules. Lysine acetylation targets protein complexes such as those containing TRIM28 and others involved in chromatin remodeling, DNA replication, transcription, posttranscriptional mRNA processing, intermediary metabolism, and protein turnover [29,32]. For example, lysine acetylation within the core histone tail domains has been shown to inhibit nucleosome formation, an effect likely related to its role in facilitating transcription. Experimentally, the effects of acetylation have been studied using recombinant mutants in which lysine residues are substituted with glutamine as a mimic of acetylated lysine, or with arginine as a mimic of unacetylated lysine. The presence of an acetylation-mimic residue within the tail domains of H2b and H4 substantially inhibits self-association and hence nucleosome formation, whereas an acetylation-mimic residue in the H3 tail affects the stability of wrapped DNA within individual nucleosomes [30,31]. These studies underscore the validity of the acetyl-lysine mimic, TRIM28-K304Q, for studying TRIM28 activities in vivo. How is the interaction between TRIM28 and KRAB-ZNFs regulated in vivo? One possibility is via the acetylation of TRIM28-K304 as we have demonstrated in our present study. Our observation of reduced binding of the KRAB domain by TRIM28-K304Q, which was not the case for either mutant K254Q, K266Q, or K289Q (mouse Trim28 K255, K267 and K290), suggests that the specific acetylation of K304 may impact the interaction between the KRAB domain. This result is consistent with a recent report showing that mouse Trim28-K305 was located in the KRAB-Trim28 interface and the K305E mutation results in a severe reduction in binding affinity with a KRAB-ZNF [38]. A number of non-ZNFs that also were specific wild-type TRIM28-interacting proteins were also identified (Appendix A), including epigenetic modifiers (EHMT1, EHMT2), protein ubiquitination proteins, RNA modification factors (YTHD2, ZCCHC4, FBRL), RNA metabolism factors (NEXT complex: MTREX, ZCCHC8, RBM7; mRNA decaying enzyme: EDC4), and a translation regulation factor (GCN1). Both wild-type and TRIM28-K304Q could associate with HP1s (CBX1, CBX3, CBX5), CENPV (centromere protein V), histones (H4, H2b2F, H2b1b; WT-TRIM28-specific: H3.3, H2a1d, H2aV; k304Q-specific: H3.1, H2a1b), RNA-binding and processing proteins (HNRNP M, HNRNP H, HNRNP F, PABP1, PABP4, REXO5, DHX9), and the transcription factor GATA1 (Appendix A). These results suggested that wild-type-TRIM28 participates in methylation of each of H3K9 and m6a RNA as well as the nuclear exosome decay pathway, whereas the K304Q mutant retains its ability to interact with HP1 and some RNA-binding proteins. In addition, we also analyzed the phosphorylation sites of wild-type and TRIM28-K304Q (Appendix A). The phosphorylated S473 and S824 were detected in K304Q mutant, indicating the phosphorylation was not affected by K304Q modification.

It was known that TRIM28 could associate with a p300-containing coactivator complex and HDAC1-containing corepressor complex to regulate skeletal muscle differentiation [39]. In the DNA repair, SIRT1-mediated deacetylation of TRIM28 stabilizes its interaction with 53BP1 [40]. Moreover, it needs to be systematically identified which acetyltransferase and deacetylase functions on K304 [28]. Furthermore, K304 is a ubiquitinated residue in our MS analysis. We also found that the protein ubiquitination complex CTLH (The C-terminal to LisH) (Appendix A) interacted with wild-type TRIM28, not TRIM28-K304Q. Based on a previous report [41], We suggest the interplay between protein acetylation and ubiquitination might control TRIM28 protein stability. The RBCC domain of TRIM28 interacts with the KRAB domain, and the C-terminal PHD and Bromodomain SUMOylation directs SETDB1 recruitment and H3K9 methylation [21,42]. We detected some acetyl residues in Bromodomain (K750. 770, 774, Appendix A), and we predicted that they might affect SUMOylation to regulate TRIM28 activity.

K562 is an erythroleukemia cell line that can be induced into erythrocytes or megakaryocytes by different drug treatments. The previous report showed that alteration of the normal balance of HAT and HDACs leads to deregulated acetylation of some proteins and thereby results in imatinib resistance [43]. We suggest that TRIM28 is one of acetylated proteins involved in the differentiation of K562 cells. In our study, TRIM28-K304Q can induce *HBE* and *ITGB3* gene expression, which is the marker of erythrocyte and megakaryocyte, respectively (Figure 3 and Figure 4). The HDAC inhibitor SAHA also would induce these genes’ expression, like the effect of TRIM28-K304Q (Figure 5a) via TRIM28 acetylation and decreasing TRIM28 targeting to these genes (Figure 5b,c). Interaction between TRIM28 and certain pluripotency-associated ZNFs such as ZNF114, ZNF483, and ZNF589/SZF1 maintains pluripotency of induced pluripotent stem cells. Disruption of any of these interactions causes the pluripotent cells to undergo differentiation [44]. In our study, wild-type TRIM28 interacts with more ZNFs than TRIM28-K304Q, including ZNF589 (Table 1). We provide evidence that compared to the weak interaction of TRIM28-K304Q, wild-type TRIM28 associates ZNF558 to suppress *ZNF568*, *ITGB3*, and transposable element *LINE1*, and associates with ZNF445 to repress *H19* (Figure 6 and Figure 7). These results are consistent with the finding that *TRIM28*-K304Q KI K562 cells undergo phenotype of differentiation, whereas the parent cells do not.

The gene expression in *TRIM28*-K304Q cells is similar to in *TRIM28*-KO cells (Figure 3 and Appendix A), both can induce many ZNFs gene expression. However, there are some differential gene expression between *TRIM28*-K304Q and *TRIM28*-KO cells (Figure 3). For example, the fibronectin was only induced in *TRIM28*-K304Q cells not in *TRIM28*-KO cells (Figure 4), which leads to their different phenotypes (Appendix A). It seems that the role of TRIM28-K304Q in gene regulation was via weak interactions with KRAB-ZNFs, which might display different effects to TRIM28-KO. It was known that TRIM28 targeted to the 3’ of KRAB-ZNF genes to repress their expression [14,15], and our results showed TRIM28 was involved in this process via interacting with other KRAB-ZNFs. The acetyl-mimic TRIM28-K304Q weakened the interaction with KRAB-ZNFs to derepress the other group of KRAB-ZNFs expression. However, the increases in KRAB-ZNFs were demonstrated by RNA-seq and qPCR, not by protein detection. We observed that the low levels of KRAB-containing proteins in *TRIM28*-K304Q cells might be due to being unstable when they did not associate with TRIM28 (Figure 1b and Figure 6b). It was consistent with the previous report [45].

The long noncoding RNA *H19* has a role in the negative regulation of body weight and cell proliferation in a mouse model [46]. In our present study, *H19* RNA was upregulated in *TRIM28*-K304Q cells. Consistent with a previous report [37], our results showed that, compared with wild-type TRIM28, TRIM28-K304Q had a weaker interaction with ZNF445, leading to upregulation of *H19* expression (Figure 7). In addition, TRIM28-mediated hypomethylation of the differentially methylated region (DMR) of H19/IGF2:IG-DMR (i.e., ICR1) might result in the decreased paternal expression of *IGF2* and increased maternal expression of *H19* [47]. In contrast to *H19*, the imprinted gene *MEG3* was downregulated in both *TRIM28*-KO and *TRIM28*-K304Q K562 cells. It is known that knockdown of TRIM28 results in differential expression of *H19* and *Gtl2* (Meg 3) in sheep fibroblasts [48]. How TRIM28-K304Q downregulates *MEG3* expression is to be further investigated.

Our present results suggest that acetylation/deacetylation of TRIM28-K304 may regulate its interaction with KRAB-ZNFs in a spatiotemporal manner in K562 cells leading to alteration of gene expression. TRIM28 plays roles in epigenetic regulation, transcriptional elongation, and protein stability modulation to control several physiological processes such as development, differentiation, energy metabolism, cell proliferation, and DNA damage response [49,50]. Particularly, TRIM28 targets endogenous retroviruses (ERVs) through KRAB-ZNFs to inhibit their expression [51], and their activation would modulate innate immunity. We suggest TRIM28 K304 acetylation might play functions in this process.

## 4. Materials and Methods

### 4.1. Cell Lines and Plasmids

Human erythroleukemic K562 cells were cultured in RPMI 1640 medium supplemented with 15% fetal calf serum (Gibco, Grand Island, NY, USA), 100 U/mL penicillin, and 0.1 mg/mL streptomycin (Gibco) at 37 °C with 5% CO_2_ in a humidified incubator. Human embryonic kidney 293T cells were cultured in Dulbecco’s modification of Eagle medium with 10% fetal calf serum, 100 U/mL penicillin, and 0.1 mg/mL streptomycin (Gibco) at 37 °C with 5% CO_2_ in a humidified incubator. The mouse FLAG-Trim28 expression plasmid was constructed as described [23]. The EcoRI-SalI fragment from FLAG-Trim28 was ligated into pCMV-HA-N (Clontech, Mountain View, CA, USA) for HA-Trim28 expression. The TRIM28 mutants were generated with the Q5 Site-Directed Mutagenesis kit (New England Biolabs, Ipswich, MA, USA) and primers shown in Appendix A. HA-ZFP57 was cloned via PCR and the sequence was confirmed. The ZNF445-, ZNF558-, and ZNF785-DDK-Myc expression plasmids were purchased from OriGene (Rockville, MD, USA). The GAL4^DBD^-KRAB expression vector and 5xGAL4^DBS^-E1bTATA-luciferase reporter were kindly obtained from Dr. Frank J. Rauscher, III.

### 4.2. CRISPR-Mediated Generation of Mutant TRIM28-K304Q in K562 Cells

Only sgRNAs with high predicted off-target scores (more precise) were selected. The *TRIM28*-K304Q mutant was produced by co-nucleofection of Cas9 RNP and a synthetic DNA repair template for homology-directed repair. Cas9 sgRNAs, targeting TRIM28, were designed using the CRISPR Design Tool at www.benchling.com. As described, the sgRNAs were synthesized by in vitro transcription and purified by denaturing PAGE (Polyacrylamide gel electrophoresis) [52]. Each purified sgRNA was refolded into a functional structure in a buffer containing 20 mM HEPES pH 7.5, 150 mM KCl, 10% glycerol, 1 mM 2-mercaptoethanol, and 1 mM MgCl_2_.A 1490-nt wild-type sequence of TRIM28, including the K304 site, was amplified by PCR using the following primer sets: TRIM28 genomic DNA (gDNA) forward primer 5′-CTCTACATCTTCCCAATAAATGGCCCAGTG-3′ and reverse primer 5′-TGTGAACAAAGCAGAACCCTCTGCCTCAGT-3′. The PCR reaction contained 200 ng genomic DNA and Taq polymerase. The PCR DNA fragment was ligated into pGEM-T Easy (Promega, Madison, WI, USA) to construct a pGEM-T Easy/hTRIM28 wild-type plasmid.

Site-directed mutagenesis was then performed to introduce the K304Q mutation in the plasmid using the primer set of the Q5 Site-Directed Mutagenesis kit (New England Biolabs) and primer sets (TRIM28-K304Q site-directed forward 5’-GCTCAATAAGCGGGGCCGTGTG and reverse 5’-TCCTGCATGATCTGCAGGATGGCC). The resulting plasmid, pGEM-T Easy/hTRIM28-K304Q, carried the K304Q (AAG>CAG) mutation and a nearby silent mutation encoding a SacI restriction sequence for screening. The DNA repair template was amplified from the plasmid by PCR using the primer sets K304Q homology-directed repair (HDR) template forward 5′-CTACCTAGCCTGACCTGCTGTG and reverse 5′-CTCACCCCAGACAGCATCATCA. Recombinant Cas9 was purified as described (DOI: 10.1002/cpmb.43). Cas9 RNP was prepared by incubating purified recombinant Cas9 and sgRNA at 1:1.2 molar ratio at 37 °C for 10 min. The DNA repair template was then added to the RNP mixture. Nucleofection of human CML K562 cells was performed in a Lonza 4D Nucleofector system using SE Cell Line 4D-Nucleofector^TM^ kit and FF-120 pulse (Lonza, Basel, Switzerland). After nucleofection, the cells were incubated at 37 °C for 48 h. A BD FACSJazz automated cell sorter (BD, Bergen County, NJ, USA) sorted single cells at the Flow Cytometry Core Facility of the Institute of Biomedical Sciences, Academia Sinica. The same primer set (TRIM28 gDNA primers) complementary to the HDR region’s flanking regions was used to amplify the target region of individual single clones. The co-integrated SacI restriction site was used to screen for the insertion of the K304Q mutation.

### 4.3. Antibodies, Preparation of Extracts, Co-IP, and Western Blotting

Cells were harvested and washed once with PBS and lysed with whole-cell extract buffer (25 mM HEPES pH 7.5, 300 mM NaCl, 1.5 mM MgCl_2_, 0.2 mM EDTA, 0.1% (*v*/*v*) Triton X-100, 1 mM DTT, protease inhibitor, and phosphatase inhibitor). Cell lysates were shaken at 4 °C for 30 min and centrifuged at 12,000× *g*, 4 °C for 5 min. The extracts were pre-cleared with protein G agarose (Sigma Aldrich, Burlington, MA, USA) for 1 h at 4 °C and immunoprecipitated with anti-FLAG M2 agarose (Sigma Aldrich) or anti-HA agarose (Sigma Aldrich) for 2 h at 4 °C. Unbound proteins were removed by washing three times with whole-cell extract buffer. SDS-PAGE sample buffer (diluted 1:4) was added to each agarose bead sample, with subsequent heating at 100 °C for 5 min, SDS-PAGE, semi-dry transfer to a polyvinylidene difluoride membrane (Millipore, Burlington, MA, USA), and Western blotting with appropriate primary antibodies and horseradish peroxidase–conjugated secondary antibodies. The immunocomplexes were visualized by enhanced chemiluminescence (PerkinElmer Life Sciences, Waltham, MA, USA) with subsequent exposure to X-ray film (FUJIFILM Corp., Tokyo, JP). The primary antibodies used were specific for the following proteins: ZNF445 and HA (Bethyl Laboratories, Montgomery, TX, USA), integrins and β-actin (Cell Signaling Technology, Danvers, MA, USA ), Gal4 (Santa Cruz Biotechnology, Dallas, TX, USA), FLAG (mouse; Sigma Aldrich), and FLAG (rabbit; [53]). Rabbit anti-human TRIM28 was generated by immunizing rabbits with peptide encompassing residues 14–43 (anti-TRIM28-N). The antibody was purified with a peptide–agarose affinity column. Monoclonal anti-human TRIM28 (clone 20A1) was produced as described [23] and obtained from Biolegend (San diego, CA, USA). The anti-acetyl-K304 antibody was produced and purified by LTK BioLaboratories (Taiwan). The secondary antibodies were goat anti-mouse (SeraCare KPL, 474-1806, Milford, MA, USA) and goat anti-rabbit (SeraCare KPL, 474-1516). The protein signals were quantitated using ImageJ [54].

### 4.4. Transfection and Luciferase Reporter Assay

The calcium phosphate precipitation transfection was used in HEK293T cells, and Lipofectamine LTX and Plus reagent (Invitrogen, Waltham, MA, USA) were used in K562 cells. HEK293T cells were seeded in 12-well culture plate and transfected with indicated plasmids and pCMV-Renilla Luc as an internal control. After 24 h, cells were harvested and lysed in passive lysis buffer (Promega). The cell lysates were subjected to dual luciferase reporter assays following the manufacturer’s protocol (Promege). The firefly luciferase activities and Renilla luciferase activities were measured by BioTek Synergy 2 (BioTek, Winooski, VT, USA) after adding substrates (LAR II and Stop & Glo^®^). The firefly luciferase activities were normalized to *Renilla* luciferase activities. The relative luciferase activities represented that the luciferase activities of reporter carrying five Gal4 DNA binding sequence were normalized to the reporter only. Each treatment group contained three duplicates, and each experiment was repeated at least three times.

### 4.5. Electrophoretic Mobility Shift Assay (EMSA)

Double stranded oligonucleotides 5′ GATCCCGGAGGACAGTACTCCGT 3′ of Gal4-binging elements were end-labeled with [γ-^32^P] ATP using T4 polyunucleotide kinase (New England Biolabs). The nuclear extracts from HEK293T cells overexpressed with FLAG-TRIM28 and GAL4^DBD^-KRAB were extracted and finally in 100 μL of buffer C (20 mM HEPES pH7.9, 400 mM NaCl, 1 mM EDTA, 1 mM EGTA, 1 mM DTT, protease and phosphatase inhibitor cocktails) [55]. The EMSA reaction was in 10 μL containing 1 μL of ^32^P-labeled probe, 2 μL of 5X binding buffer (50 mM HEPES pH7.9, 250 mM KCl, 1 mM EDTA, 12.5 mM DTT, 50% glycerol, 0.25% NP40), 2 μL of nuclear extracts, and 0.3 μg of poly(dI-dC). After incubation at room temperature for 30 min, 1 μg of anti-Gal4 or anti-FLAG was added and incubated at room temperature for further 1 h. Then, 0.5 μL of 0.05% BPB was added and the mixes were loaded to 5% native gel in TG buffer (25 mM Tris, 192 mM glycine) at 150 V for 50 min. Baking at 80 °C for 30 min and expose at −80 °C freezer for 4–16 h depending on the signal intensity.

### 4.6. Identification of TRIM28 and TRIM28-K304Q Associated Proteins

Peptide identification by MS was performed by the Mass Spectrometry Common Facility at the Institute of Biological Chemistry, Academia Sinica, using an LTQ-Orbitrap Velos system (Thermo Fisher Scientific, Waltham, MA, USA). Data interpretation and correlations between the spectra and amino acid sequences within a human EST database and customized FLAG-Trim28 sequence were analyzed using Mascot (Matrix Science software package).

### 4.7. RNA-seq

Libraries were prepared for next-generation sequencing according to the manufacturer’s protocol (Agilent Technologies, Santa Clara, CA, USA). Total cellular RNA was extracted using TRIzol Reagent (Invitrogen). The total RNA of each sample was quantified with a 2100 Bioanalyzer (Agilent Technologies), NanoDrop (Thermo Fisher Scientific), and 1% agarose gel electrophoresis. Total RNA (1 μg) with an RNA integrity value > 6.5 was used for library preparation. Poly(A) mRNA was isolated using the Poly(A) mRNA Magnetic Isolation Module or rRNA removal kit (New England Biolabs). The mRNA was fragmented and primed using First Strand Synthesis Reaction Buffer (New England Biolabs) and random primers. First-strand cDNA was synthesized using ProtoScript II Reverse Transcriptase (New England Biolabs), and second-strand cDNA was synthesized using the Second Strand Synthesis Enzyme Mix (New England Biolabs). Bead-purified double-stranded cDNA was then treated with End Prep Enzyme Mix to repair both ends, and dA-tailing was carried out in the same reaction, followed by T-A ligation to add adaptors to both ends. Size selection of adaptor-ligated DNA was then performed using beads, and fragments of ~420 bp (insert size ~300 bp) were recovered. Each sample was then amplified by PCR for 13 cycles using primers P5 and P7, each of which carried sequences that could anneal with the flowcell to perform bridge PCR; P7 carried a six-base index to allow multiplexing. The PCR products were cleaned up using beads, validated using a Qsep100 (Bioptic), and quantified with a Qubit3.0 fluorometer (Invitrogen). Then, libraries with different indices were multiplexed and loaded on an Illumina HiSeq instrument (Illumina). Sequencing was carried out using a 2’ 150 bp paired-end configuration, and image analysis and base calling were conducted with HiSeq Control Software (HCS)+ OLB +GAPipeline-1.6 (Illumina) on the HiSeq instrument.

### 4.8. Lentivirus-Mediated Gene Knockdown

Lentiviral vector–mediated short hairpin RNA technology was used for *ZNF445* and *ZNF558* knockdown. The short hairpin RNA target sequences, GCGCTATAAATGTAATCTATG (TRCN0000415362) and ATCAAACTTTACTCGTCATAT (TRCN0000435973) to *ZNF445*, CGTGGAGTGAAACTCAATGAA (TRCN0000019403) to *ZNF558*, and scrambled control pLKO.1-shLuc (TRCN0000072243) was obtained from the National RNAi Core Facility at Academia Sinica. Viruses were produced using calcium phosphate–mediated transfection. Early subcultures of 293T cells were cotransfected with 14 μg pPGK-GFP, pLKO.1-shLuc, or pLKO.1-shZNF445 or pLKO.1-shZNF558 and 14 μg pCMVΔR8.91 and 2 μg pMD.G. After 8 h, the medium was replaced with K562 cell maintenance medium to collect virus. Then, K562 cells in a 6-well plate were infected with viral supernatants in the presence of 8 μg/mL polybrene for 48 h and further selected with 3 μg/mL puromycin for one week. Knockdown efficiency was determined based on RT-qPCR and Western blotting.

### 4.9. Real-Time PCR

K562 cells in a 6 cm dish were harvested, and total RNA was prepared with TRIzol reagent (1 mL per dish). After quantification by measuring absorbance at 260 and 280, 2 μg RNA was treated with DNase I (Invitrogen) and then reverse transcribed into cDNA using Superscript IV (Invitrogen). qPCR was performed with the Corbett Research RG-6000 Real-Time PCR Thermocycler (Qiagen, Hilden, Germany). The total volume was 20 μL, including QuantiNova SYBR Green master mix (Qiagen), 20-fold diluted cDNA, and 0.3 μM each of the forward and reverse primers (Appendix A). The amplification conditions were 60 cycles of 95 °C for 10 s and 60 °C for 15 s. The results were analyzed by the 2^–∆∆Ct^ relative quantitation method. All experiments were independently repeated three times.

### 4.10. Chromatin Immunoprecipitation (ChIP)

K562 cells were cultured to a density of 2 × 10^6^/mL and harvested for cross-linking with 1% formaldehyde in a medium at room temperature for 10 min. Glycine was added (final concentration, 125 mM) to quench unreacted formaldehyde for 5 min. Cells were pelleted by centrifugation, and the pellet was washed with 5 mL cold PBS. Cells (2 × 10^7^) were lysed in 400 μL lysis buffer (5 mM HEPES pH 8.0, 85 mM KCl, 0.5% (*v*/*v*) NP-40, and protease inhibitor cocktail) on ice for 15 min, and the cell suspension was mixed gently every 5 min. Nuclei were collected by centrifugation at 9000× *g* for 5 min at 4 °C, resuspended in 200 μL nuclear lysis buffer (50 mM Tris-HCl pH 8.0, 10 mM EDTA, 1% SDS, and protease inhibitor cocktail), and kept on ice for 10 min. The nuclear lysates were sonicated using the middle setting of an ice-water Bioruptor (Diagenode, Liege, Belgium) for 10–15 min total (30 s on, 30 s off). The resulting sheared chromatin (2 µL) was resolved by electrophoresis through a 2% agarose gel to check that the length of DNA fragments was 200–600 bp. Each sheared chromatin sample was centrifugated at 12,000× *g* for 5 min at 4 °C. The supernatant was diluted 10-fold with buffer containing 20 mM Tris-HCl pH 8.1, 167 mM NaCl, 0.01% SDS, 1% Triton X-100, 1 mM EDTA, and protease inhibitors. Each chromatin complex sample was divided into two equivalent volumes for IP with either normal IgG or anti-TRIM28-N. After incubating with the antibody at 4 °C overnight, protein A/G magnetic beads (20 µL) was added for an additional 2 h. The beads were washed 1 time each with 0.5 mL of each of the following buffers: low-salt wash buffer (0.1% SDS, 1% Triton X-100, 2 mM EDTA, 20 mM Tris-HCl pH 8.1, 150 mM NaCl), high-salt wash buffer (0.1% SDS, 1% Triton X-100, 2 mM EDTA, 20 mM Tris-HCl pH 8.1, 500 mM NaCl), LiCl wash buffer (0.25 M LiCl, 1% (*v*/*v*) IGEPAL CA630, 1% deoxycholic acid–sodium salt, 1 mM EDTA, 10 mM Tris pH 8.1), and TE buffer (10 mM Tris-HCl pH 8.0, 1 mM EDTA). Each wash step was done for 5 min on a rotating platform at 4°C. Chromatin complexes were eluted with 100 μL elution buffer (1% SDS, 0.1 M NaHCO_3_, containing 1 μL proteinase K) for 2 h at 62 °C with mixing, with subsequent incubation at 95 °C for 10 min to de-crosslink the chromatin complexes. Samples were cooled to room temperature, and the supernatant was transferred to a new tube. After extraction by phenol/chloroform, the DNA was precipitated by ethanol and analyzed by qPCR with specific primers (Appendix A).

### 4.11. Statistical Analysis

All data are presented as the mean ± SD of at least three independent experiments. Statistical significance (* *p* < 0.05, ** *p* < 0.01, *** *p* < 0.001) was determined by the one-tailed Student’s *t*-test.

## 5. Conclusions

TRIM28 is highly expressed in cells to silence gene expression in heterochromatin; especially, TRIM28 associates with KRAB-ZNFs to inhibit transposable elements. Its activity can be modulated by post-translational modifications such as phosphorylation and SUMOylation. This study uses the acetyl-mimic mutant in biochemical and functional assays to demonstrate that K304 acetylation decreases interaction with KRAB-ZNFs and leads to some differentiation-related gene up-regulation. This is the first report to investigate the acetylation regulation on TRIM28, and it provides evidence to display the dynamic interaction between KRAB-ZNF and TRIM28.

## Figures and Tables

**Figure 1 ijms-24-09830-f001:**
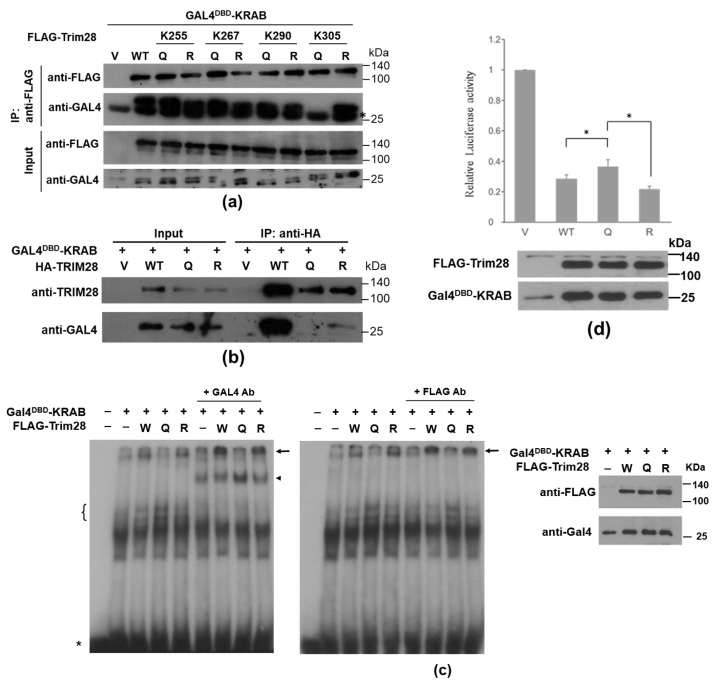
Impaired interaction between mouse Trim28-K305Q or human TRIM28-K304Q and KRAB domain-containing proteins. (**a**) Co-IP analysis of FLAG-tagged mouse Trim28-WT (wild-type), -K255Q or R, -K267Q or R, -K290Q or R, and K305Q or with Gal4^DBD^-KRAB in TRIM28-knockout (KO) K562 cells. The protein complexes were IPed by anti-FLAG and analyzed by Western blotting with antibodies against FLAG and Gal4. The asterisk indicates non-specific signals. (**b**) Co-IP of Gal4^DBD^-KRAB with HA-tagged human TRIM28-WT, -K304Q, or -K304R in K562 cells. (**c**) TRIM28 targets on DNA through KRAB-DNA-binding protein. EMSA was conducted using nuclear extracts from HEK293T cells transfected with Gal4^DBD^-KRAB and FLAG-Trim28 (WT or mutant Trim28 K305Q (Q) or K305R (R)) expression plasmids as indicated. Nuclear extracts were incubated with ^32^P labeled Gal4-binging elements with and without addition of anti-Gal4 antibody (left panel) or anti-FLAG antibody (middle panel). The asterisk indicates probe position; the bracket indicates Gal4^DBD^-KRAB and DNA complexes; the arrowhead indicates Gal4^DBD^-KRAB-DNA complexes shifted by anti-Gal4; the arrows indicate Gal4^DBD^-KRAB-Trim28-DNA complexes shifted by antibodies. All nuclear extracts from 293T cells were monitored by Western blot analysis (right panel). (**d**) Luciferase reporter assay by cotransfecting HEK293T cells with 5xGal4^DBS^-E1bTATA-driven luciferase and Gal4^DBD^-KRAB and FLAG-Trim28 expression plasmids. * *p* < 0.05. Lower panel: Western blotting with antibodies against FLAG and Gal4.

**Figure 2 ijms-24-09830-f002:**
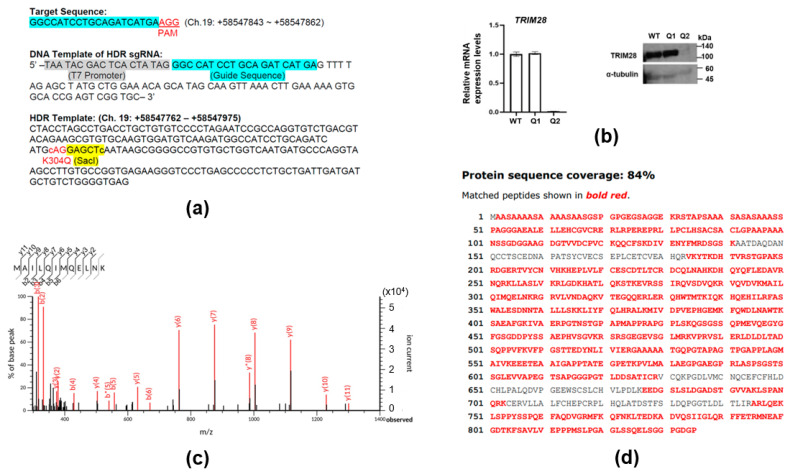
Generation and characterization of *TRIM28*-K304Q knock-in (KI) K562 cells. (**a**) Sequences of target, sgRNA, and homology-directed repair (HDR) donor DNA for *TRIM28*-K304Q knock-in. The blue highlighted the guide sequence; the grey highlighted the T7 promoter; the yellow highlighted the SacI digested sequence. The PAM and K304Q sequences were indicated by red. (**b**) TRIM28 mRNA and protein expression in KI clones Q1 and Q2. RNAs and proteins were isolated from wild-type (WT) and *TRIM28*-K304Q KI clones for qPCR analysis and Western blotting with anti-TRIM28 and internal control anti-α-tubulin, respectively. (**c**) MS analysis of TRIM28-K304Q proteins. A 304Q-containing peptide (residues 297–308, MAILQIMQELNK) was identified. The b* and y* indicate peaks are seen for ions that have lost ammonia (−17 Da). (**d**) Amino-acid sequence confirmed by MS.

**Figure 3 ijms-24-09830-f003:**
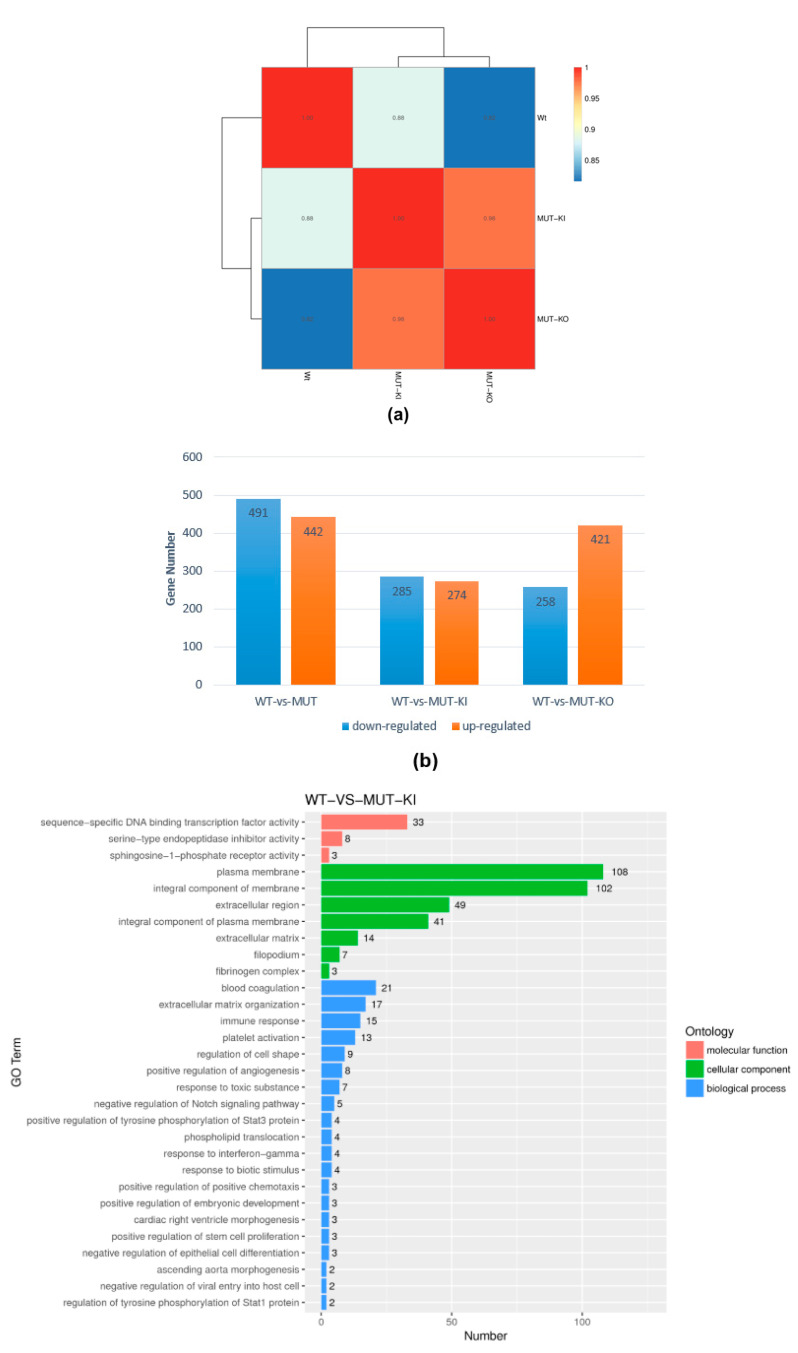
RNA-seq analysis. (**a**) RNA-seq correlations. The Pearson correlation coefficient indicates the degree of linearity between two variables. The greater the absolute value (ranging between 0 and 1), the stronger the linear relationship. Wild-type (WT): K562; MUT-KI: *TRIM28*-K304Q KI; MUT-KO: *TRIM28*-KO. (**b**) Bar graph of genes significantly up- or down-regulation between groups. MUT: *TRIM28*-K304Q KI + *TRIM28*-KO. (**c**) GO enrichment histogram. The number of differentially expressed genes in each GO term is shown along with the specification of the relevant biological process, cellular component, or molecular function. Shown are the top 30 most prominent GO categories. Upper: number of differentially expressed genes in each GO category; lower: *p*-value histogram. (**d**) RT-qPCR assays verified the identity of certain ZNFs, imprinting genes, and transcription factors in WT and TRIM28-K304Q KI cells. * *p* < 0.05, ** *p* < 0.01, *** *p* < 0.001; ns, not significant.

**Figure 4 ijms-24-09830-f004:**
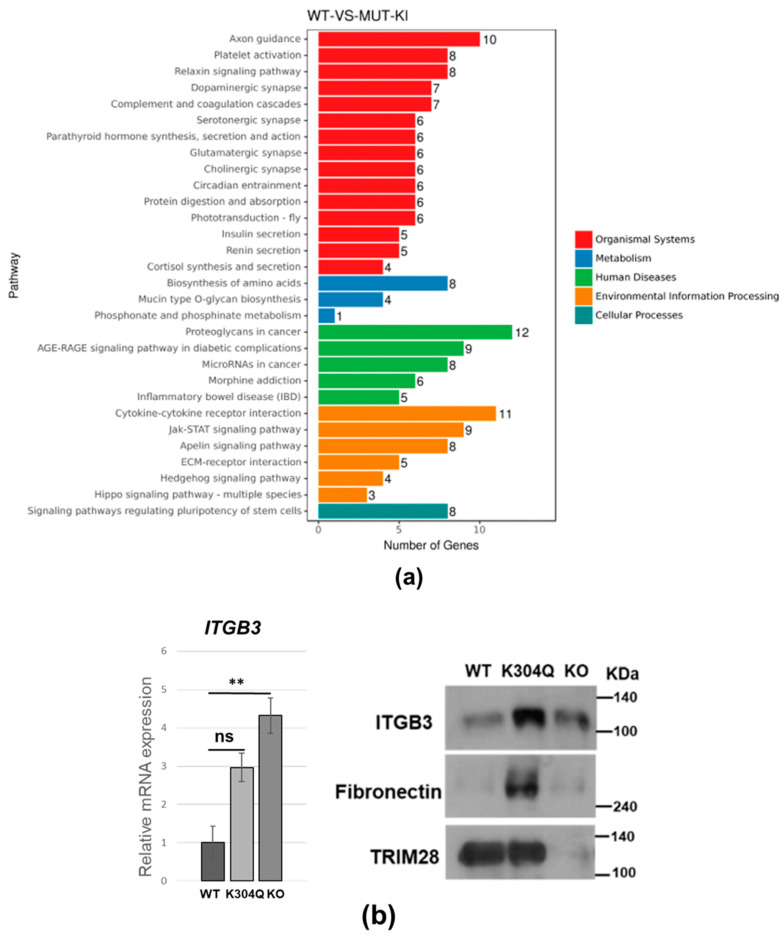
TRIM28-mediated pathways. (**a**) KEGG enrichment histogram. x axis: gene number. y axis: pathway term. (**b**) Integrin β3 expression in wild-type (WT), *TRIM28*-K304Q, and *TRIM28*-KO K562 cells. (**c**) qPCR analysis of beta globin gene expression in wild-type (WT), K304Q and *TRIM28*-KO cells. Total RNA from K562 cells was isolated and used for RT-qPCR to assess the expression of HBE and HBG. ** *p* < 0.01, *** *p* < 0.001, ns, no significance, *p* = 0.07.

**Figure 5 ijms-24-09830-f005:**
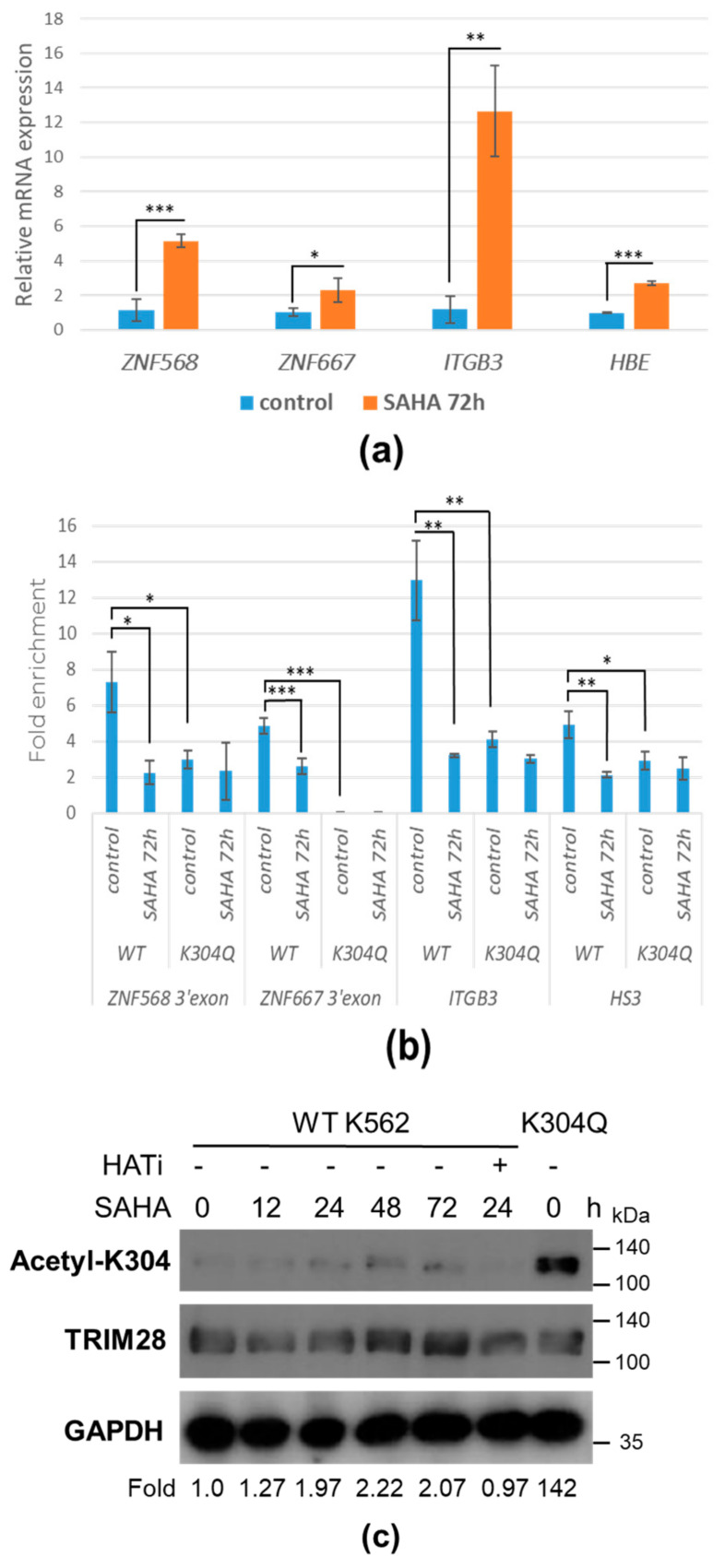
HDAC inhibitor SAHA modulates gene expression via TRIM28-K304 acetylation. (**a**) K562 cells were treated with 1 μM SAHA for 72 h and RNA expression levels of *ZNF568, ZNF667, ITGB3* and *HBE* were analyzed by RT-qPCR. (**b**) K562 cells were treated with 1 μM SAHA for 72 h and ChIP assays were performed with anti-TRIM28-N. The precipitated DNAs were PCR analyses with primers for *ZNF568* 3’UTR, *ZNF667* 3’UTR, *ITGB3*, and beta-globin locus control region HS3. (**c**) K562 cells were treated with 1 μM SAHA and 10 μM HAT inhibitor Garcinol for the indicated time intervals. The cell extracts were Western blotted with anti-aceyl-K304, anti-TRIM28, and anti-GAPDH. The acetylated levels of K304 were quantitated using ImageJ. The signal of acetyl-K304 was normalized with the levels of TRIM28 and GAPDH. * *p* < 0.05, ** *p* < 0.01, *** *p* < 0.001.

**Figure 6 ijms-24-09830-f006:**
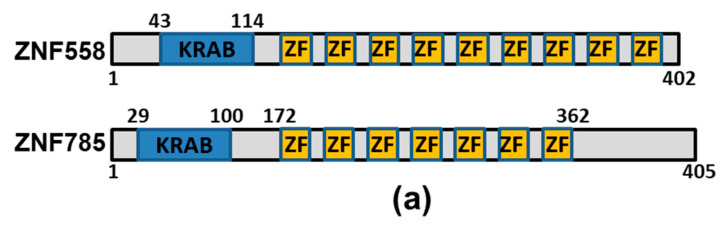
TRIM28-K304Q regulates gene expression through altering interaction with KRAB-ZNF proteins. (**a**) Schematic presentation of ZNF558 and ZNF785, containing one KRAB domain and nine zinc-finger (ZF) and seven ZFs, respectively. (**b**) Co-IP of Myc-Flag-ZNF558 or-ZNF785 with HA-tagged human TRIM28-WT, -K304Q, or -K304R in HEK293T cells. The asterisks indicate non-specific signals. The IPed levels of ZNF785 were normalized with input levels of ZNF785 and TRIM28 constructs using ImageJ. The IPed ratio was indicated. (**c**) RNA expression levels of *ZNF558*, *ZNF568*, *ITGB3*, and *LINE1* in control (shLuc) and ZNF558 knocked-down (shZNF558) K562 cells. *** *p* < 0.001.

**Figure 7 ijms-24-09830-f007:**
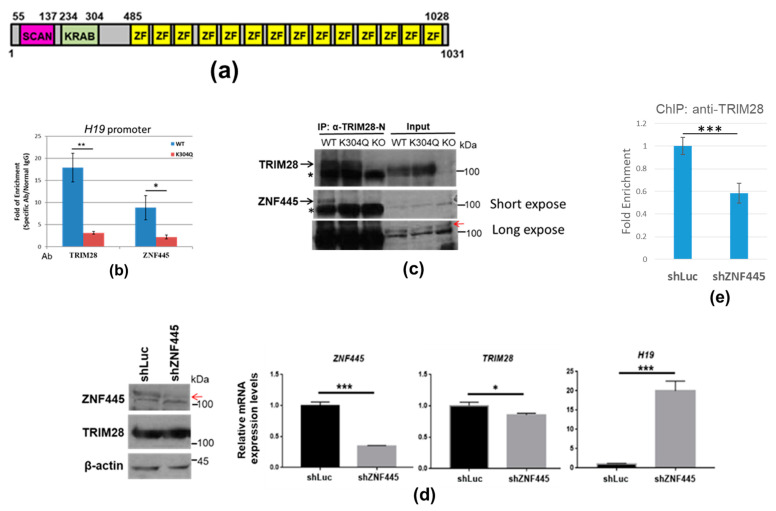
Interaction between TRIM28 and ZNF445 inhibits *H19* gene expression. (**a**) Schematic presentation of ZNF445. It contains SCAN domain, KRAB domain, and 14 Zinc fingers. (**b**) ChIP by anti-TRIM28 and anti-ZNF445. The precipitated DNA was analyzed with *H19* promoter in wild-type (WT) and *TRIM28*-K304Q K562 cells. (**c**) Immunoprecipitation analysis in wild-type, *TRIM28*-K304Q KI, and *TRIM28*-KO K562 cells. The protein complexes were precipitated by anti-TRIM28-N and Western blotted with anti-ZNF445. The asterisk indicates a non-specific signal. The red arrow indicates the specific ZNF445 band. (**d**) Knockdown of ZNF445 in K562 cells increases *H19* expression. Left panel: Western blotting confirmed ZNF445 knockdown. The red arrow indicates the specific ZNF445 band. Right panels: qPCR analysis using primers of *ZNF445, TRIM28* and *H19*, respectively. (**e**) Knockdown of ZNF445 in K562 cells decreases TRIM28 enrichment at the *H19* promoter. * *p* < 0.05, ** *p* < 0.01, *** *p* < 0.001.

**Table 1 ijms-24-09830-t001:** The ZNFs interacted with wild-type TRIM28 and TRIM28-K304Q.

ZNFs associated with wild-type TRIM28 only	ZNF7	ZNF12	ZNF17	ZNF20	ZNF22	ZNF25	ZNF33A	ZNF33B
	ZNF41	ZNF44	ZNF45	ZNF57	ZNF84	ZNF101	ZNF107	ZNF121
	ZNF136	ZNF181	ZNF184	ZNF197	ZNF208	ZNF211	ZNF227	ZNF229
	ZNF235	ZNF257	ZNF264	ZNF311	ZNF317	ZNF320	ZNF324B	ZNF354B
	ZNF383	ZNF419	ZNF440	ZNF468	ZNF480	ZNF490	ZNF493	ZNF512
	ZNF529	ZNF543	ZNF550	ZNF552	ZNF554	ZNF555	ZNF558	ZNF563
	ZNF564	ZNF567	ZNF569	ZNF583	ZNF587B	ZNF589	ZNF607	ZNF611
	ZNF615	ZNF620	ZNF624	ZNF644	ZNF655	ZNF678	ZNF684	ZNF697
	ZNF699	ZNF701	ZNF707	ZNF718	ZNF721	ZNF726	ZNF749	ZNF766
	ZNF776	ZNF778	ZNF780A	ZNF785	ZNF789	ZNF791	ZNF799	ZNF805
	ZNF813	ZNF841	ZNF845	ZFP1	ZFP82	ZBT11	ZKSC1	ZKSC8
ZNFs associated with wild-type and K304Q TRIM28	ZNF8	ZNF34	ZNF74	ZNF77	ZNF90	ZNF91	ZNF92	ZNF93
	ZNF100	ZNF124	ZNF160	ZNF195	ZNF250	ZNF253	ZNF267	ZNF273
	ZNF274	ZNF316	ZNF324A	ZNF354A	ZNF417	ZNF430	ZNF432	ZNF441
	ZNF445	ZNF460	ZNF485	ZNF486	ZNF561	ZNF562	ZNF566	ZNF587
	ZNF595	ZNF614	ZNF627	ZNF649	ZNF669	ZNF670	ZNF680	ZNF688
	ZNF689	ZNF708	ZNF724	ZNF728	ZNF736	ZNF738	ZNF764	ZNF792
	ZNF808	ZNF816	ZNF823	POGK	ZFP92	RBAK		
ZNFs associated with K304Q TRIM28 only	ZNF98	ZNF140	ZNF182	ZNF626	ZNF681	ZNF737	ZNF878	

## Data Availability

Data are contained within the article or Appendix A.

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
