# Peer review of "Acetylation-Mimic Mutation of TRIM28-Lys304 to Gln Attenuates the Interaction with KRAB-Zinc-Finger Proteins and Affects Gene Expression in Leukemic K562 Cells"

_ijms, 2023, doi:10.3390/ijms24129830_

Round 1

Reviewer 1 Report

The manuscript presents an interesting and well-conducted study on TRIM28 regulation by acetylation. The manuscript is well organized, structured and the experimental procedures are done properly with proper controls. The topic is interesting, and the results presented here are novel – as far as I could perceive, no articles on TRIM28 acetylation are known. However, the in its current form, the article, could benefit from some revisions and additional experiments. Specifically, two important topics raised some questions:

i)        Is it possible that the acetylation on lysine 304 is regulating TRIM28 stability rather than activity? Is it known that acetylation of lysine residues could induce stability by competing directly with lysine ubiquitination – the authors should address this concern.

ii)       The title needs to be revised or the authors should perform additional experiments showing that TRIM28 acetylation on lysine 304 could in fact induce cellular differentiation of K562 cells – the authors only show differences in the gene expression profile.

These revisions would help to strengthen the study and its findings and provide a more complete understanding of TRIM28 acetylation. Overall, the article shows great promise and has the potential to make a significant contribution to the field with the suggested revisions.

Author Response

Please see the PDF file

Reviewer 2 Report

This manuscript aims to determine if acetylation of TRIM28 affects its function, specifically its binding to Kruppel-associated box zinc-finger proteins (KRAB-ZNFs). They make both acetyl-mimic and non-acetylatable mutants at different lysine residues in TRIM28 that are known to be acetylated and find that only an acetylation mimic at lysine 304 (K304Q) alters interaction with KRAB-ZNFs. They then do RNAseq analysis of WT K562 cells, TRIM28 KO K562 cells, and K562 cells that have CRISPR-mediated knock-in of TRIM28-K304Q. In the knock-in line, they see increased expression of differentiation-related genes, zinc-finger-protein genes, and imprinting genes. The authors conclude that acetylation of K304 is a switch for regulating its interaction with KRAB-ZNFs.

The findings in this paper are novel, though some experiments could use further quantification or better controls. I recommend the paper for publication with the following revisions.

1.)  Figure 5C uses an antibody the authors have generated against acetyl-K304 for this paper. There is no characterization of this antibody provided. Since it has not been published before, they need to confirm that the antibody is specific for K304 acetylation and doesn’t recognize unacetylated TRIM28. This could be done using recombinant protein. Also, the antibody appears to recognize K304Q, which doesn’t make sense. While a glutamine residue may chemically mimic acetylation, it should not be recognized as acetylation by an acetylation-specific antibody.

2.)  Figure 6B is used to claim that K304Q interacts less with ZNF785 than WT TRIM28 or TRIM28-K304R. However, the difference between K304Q and K304R is very small. Quantification of at least three separate experiments is needed here to demonstrate that there is actually a statistically significant difference between K304Q and K304R. This is an important point, because otherwise the presence or absence of acetylation is working in the same way and this contradicts the main hypothesis of the paper.

3.)  Whole discussion section is just a summary of the results. This space would be better used by explaining the relevance of their current data. 

Overall English fine, small minor corrections needed.

Author Response

Please see the PDF file

Reviewer 3 Report

This report is an extension of previous work done by this group, characterizing the transcriptional co-repressor TRIM28-regulated gene expression in K562 erythroleukemia cells. Here, Chang and colleagues study acetylation's impact on the TRIM28 activity. Using a mass spectrometry analysis, the authors found several lysine residues in the coiled-coil domain of TRIM28  were acetylated. Of those sites,  the authors showed that acetylation-mimic mutant TRIM28-K304Q has a reduced interaction with KRAB-ZNFs compared to the WT protein. The authors then created a TRIM28-K304Q knock-in in K562 cells by the CRISPR-Cas9 gene editing approach. Transcriptomic analysis showed that the expression of TRIM28-K304Q resulted in an altered global expression profile from wild-type K562 cells, including the loss of repression and consequent upregulation of several gene targets, that wild-type TRIM28 inhibited via binding with KRAB-ZNFs These results suggested that TRIM28-K304Q mutant alters interaction with KRAB-ZNFs to modulate gene expression. Furthermore, expression of TRIM28- K304Q mutant also led to the upregulation of cell differentiation-related genes, HBE and ITGB3. The authors conclude that acetylation of TRIM28 in K304 residue promotes cell differentiation in leukemic K562 cells.

 Major comments

 1. Intriguingly, the alteration in the acetylation of only one residue ( K304) in the RBCC domain significantly impacts TRIM28 activities, showing no signs of compensation from the other lysine residues in this domain. While the constitutive and high acetylation levels of K304 residue (see point 2) offers an attractive model to study the regulation and activities of TRIM2, the results could be more convincing. The study would gain support if the authors did several experiments (gene expression, protein interaction profile) in an acetylation defective K304R knock-in the cell line.

 2. At basal levels (Figure 5c, no SAHA treatment), the acetylation levels in K304 residue in the TRIM28-K304Q knock are much higher than in the WT cells. Do these levels are biologically significant? It will be interesting to evaluate the impact of the acetylation levels of other lysine residues in TRIM28-K304Q cells. The effect of SAHA treatment on the acetylation of K304 residue in WT cells is minimal (Figure 5c); a densitometry analysis should be included.

 3. While MS indicated that in TRIM28-K304Q cells, several phosphorylation residues were detected, the phosphorylation levels in key residues (i.e., Ser473 and Ser824) cannot be assessed accurately. Therefore, the authors should include a western blot to address possible changes in the phosphorylation levels in TRIM28-K304Q cells.

4. SUMOylation of TRIM28 is important for its activities. How does the acetylation of the lysine residues in the RBCC domain of TIRM28 impact its SUMOylation status? Does the acetylation /deacetylation of K304 affect the TRIM28 SUMOylation?

 5. By transcriptomics analysis, the authors conclude that acetylation mimics mutation in TRIM28 K304 residue impacts cell differentiation in K562 cells. As presented in Figures 3 and 4, TRIM28-K304Q can induce HBE and ITGB3 gene expression, which are cell differentiation markers. These conclusions could be strengths if they are validated experimentally. For example, treatment of all the  K562 cell lines with butyrate to determine the erythroid differentiation to assess the hemoglobin synthesis by hemoglobin staining of K562 cells with O-dianisidine.

 6. As an important internal and comparison control, the expression of ITGB3 (Figure 4b) and the genes validated in Figure 3d in TRIM28-KO K562 cells should be presented.

 7. The discussion needs to be improved. Which are the potential deacetylase(s) and acetyltransferase(s) that modify KAP1? Based on the results present, do the authors expect that a specific deacetylase and acetyltransferase could mediate acetylation status K304? In addition to modulating pluripotency, authors should consider discussing their results and the potential impact of the acetylation in TRIM28 in the context of another biological process involving this protein.

8. Besides HP1 interaction, did the TRIM28-K304Q protein retain its interaction with other known WT partners (for example, NuRD complex proteins, DNMTS, H3K9me3, the H3K9me3 methyltransferase SETDB1, G9a methyltransferase)?

 Minor comments

1.   Figure S3B legend. Describe for how long the cells were SAHA and Imatinib treatments.

2. Figure 3B shows the far-left comparison ( WT vs MUT). What is MUT?

3. How similar are the mouse and human TRIM28 proteins?

4. From the protein-protein interaction analysis. Do all the ZNFs interacting with TRIM28-K304Q present a KRAB domain? Is there something unique in the 7 ZNFs that only associate with K304Q TRIM28?

Author Response

Please see the PDF file

Round 2

Reviewer 1 Report

The manuscript presents an interesting and well-conducted study on TRIM28 regulation by acetylation. The manuscript is well organized, structured and the experimental procedures are done properly with proper controls. The topic is interesting, and the results presented here are novel – as far as I could perceive, no articles on TRIM28 acetylation are known. The authors addressed the concerns raised in the first revision very satisfactory.

Overall, and as stated in the previous revision, the article provides significant contributions to the field.

Reviewer 3 Report

The authors have covered satisfactorily the revision comments.

No further comments